# Geophysical constraints on the reliability of solar and wind power worldwide

Dan Tong [1,2,3✉], David J. Farnham [3], Lei Duan [3], Qiang Zhang [1], Nathan S. Lewis[3,4], Ken Caldeira [3,5] & Steven J. Davis [2,3,6]

If future net-zero emissions energy systems rely heavily on solar and wind resources, spatial and temporal mismatches between resource availability and electricity demand may challenge system reliability. Using 39 years of hourly reanalysis data (1980–2018), we analyze the ability of solar and wind resources to meet electricity demand in 42 countries, varying the hypothetical scale and mix of renewable generation as well as energy storage capacity. Assuming perfect transmission and annual generation equal to annual demand, but no energy storage, we find the most reliable renewable electricity systems are wind-heavy and satisfy countries' electricity demand in 72–91% of hours (83–94% by adding 12 h of storage). Yet even in systems which meet >90% of demand, hundreds of hours of unmet demand may occur annually. Our analysis helps quantify the power, energy, and utilization rates of additional energy storage, demand management, or curtailment, as well as the benefits of regional aggregation.

[1] Ministry of Education Key Laboratory for Earth System Modeling, Department of Earth System Science, Tsinghua University, Beijing 100084, China. [2] Department of Earth System Science, University of California, Irvine, CA 92697, USA. [3] Department of Global Ecology, Carnegie Institution for Science, Stanford, CA 94035, USA. [4] Division of Chemistry and Chemical Engineering, California Institute of Technology, Pasadena, CA 91125, USA. [5] Breakthrough Energy, 4110 Carillon Pt., Kirkland, WA 98033, USA. [6] Department of Civil and Environmental Engineering, University of California, Irvine, CA 92697, USA. ✉email: dantong@tsinghua.edu.cn

Stabilizing mean global temperatures requires a global transition to energy systems with near-zero (or net-negative) carbon dioxide equivalent emissions[1–3]. In cost-optimized scenarios that accomplish this transition, solar and wind resources often supply a large share (e.g., >60%) of electricity[4–10]. Designing and operating a highly reliable electricity system that is dependent on such large shares of wind and solar generation can be a challenge, however, due to the variable and uncertain nature of solar and wind resources[11,12]. The efficacy of meeting electricity demands with generation from solar and wind resources depends on factors such as location and weather; the area over which generating assets are distributed; the mix and magnitude of solar and wind generation capacities; the availability of energy storage; and firm generation capacity[11–16]. Meanwhile, reliability standards in industrialized countries are typically very high (e.g., targeting <2–3 h of unplanned outages per year, or ~99.97%[17]). Resource adequacy planning standards for "1-in-10" are also high: in North America (BAL-502-RF-03)[18], generating resources must be adequate to provide no more than 1 day of unmet electricity demand—or in some cases 1 loss of load event—in 10 years (i.e., 99.97% or 99.99%, respectively)[19].

Here, we present a systematic analysis of the ability of specified amounts of solar and wind generation to meet electricity demands in 42 major countries across a range of assumptions associated with transmission, energy storage, and generation amounts. In particular, we assess spatial and temporal gaps between electricity demand and the availability of solar and wind resources, which represent gaps that must be filled by other non-emitting generation technologies or operating strategies in reliable electricity systems based on zero-carbon sources. The complementarity of renewable energy sources for this study is defined as a hybridization of solar-wind resources over a given area (here, countries), which we estimate by the Kendall correlation coefficient of these resources across 39-years of resource data[20]. Our goal is to identify the opportunities, complementarity, and challenges of variable renewable resources in greater detail than can be done by integrated assessment models that have multi-year time steps. Our results do not account for realistic power system specifications. Rather, we examine fundamental geophysical constraints on wind- and solar-dominated power systems independent of cost estimates. Note that we do not mean to suggest that the temporal variability of such resources would ever make it physically impossible to meet a given electricity demand (with enough capacity the solar and wind resources would be able to meet demand), but rather the extent to which such variability may determine the economic or socio-political feasibility of reliable systems. Our results will thus continue to be informative even as technological and socio-political feasibility evolves.

Details of our analytical approach are in the "Methods" section. In summary, we use 39 years (1980–2018) of gridded (0.5° × 0.625°) and hourly reanalysis data[21,22] and actual/projected hourly electricity demand from a single recent year to evaluate the adequacy of solar and wind resources to meet electricity demand in each of 42 major countries (data sources and countries are listed in Supplementary Data 1). First, hourly, area-weighted capacity factors for both solar and wind resources are calculated over each country (or region), assuming perfect transmission within the country or region. Then we exogenously specify (1) the mix of solar and wind generation, (2) the overall level of annual generation from these sources, and (3) the capacity of energy storage, and analyze the ability of the specified technologies to meet hourly demand. We analyze systems ranging from 100% solar (no wind) to 100% wind (no solar), in which total annual generation ranges from equal to annual demand ("1x generation") to up to three times annual demand ("3x generation"), and in which available energy storage ranges from none

("0 h") to 12 h of mean demand ("12 h"). In addition, we simulate the impacts of different demands (i.e., demand load profiles) and technologies (i.e., single-axis and dual-axis solar tracking systems) on electricity system reliabilities as sensitivity tests. The number of countries, years of reanalysis data, and different system configurations we analyze require computation and analysis of ~300,000 year-long simulations.

## Results

**Resources and demand variability.** Figure 1 shows the seasonal and daily variability of solar and wind resources and electricity demand in the six countries with the greatest electricity demand on every continents except Antarctica (results from six other major countries and continent-level aggregated regions are shown in Supplementary Figs. 1 and 2, respectively). Solar and wind consistently peak in summer and winter, respectively, in countries of the Northern Hemisphere (seasons are reversed in countries of the Southern Hemisphere; Fig. 1a–f). The seasonal cycles of solar and wind thus suggest potential complementarity in many countries (e.g., China, Fig. 1a; and Germany, Fig. 1b). However, during the 39-year period, interannual variability of wind is consistently much greater than that of solar in most countries (Fig. 1a–f), though the magnitude of these resources' variability differs substantially between two particular countries. For example, Germany's small area (0.36 million km$^2$) and high latitude (centroid 51.2 °N) result in large interannual variations in both solar (measured by the robust coefficient of variation[23]; RCoV = 58.8%) and wind resources (RCoV = 47.2%, Fig. 1b), whereas solar resource variability is very low (RCoV = 6.6%) in the larger and tropical country of Brazil (8.52 million km$^2$ and centroid 14.2 °S; Fig. 1e). Wind resources are also more variable than solar resources on the time scale of days to weeks in each country, which acts to limit and undermine the resources' seasonal complementarity. Electricity demand profiles for each country are determined by factors such as economic conditions, prevailing weather conditions and consumer usage patterns[24]. Therefore, electricity demand for two countries can have unique seasonal shapes and a range of variabilities even if they have similar wind and solar resources. For example, seasonal variability of demand in France (RCoV = 14.4%; Supplementary Fig. 1e) is greater than that in Germany (RCoV = 7.4%; Fig. 1b), despite the countries' similar wind and solar resource profiles.

Daily cycles of solar and wind resources in each country are also somewhat complementary. Wind power usually peaks at night and rarely falls to zero when resources are aggregated over an entire country. This daily cycle is not substantially different during the summer and winter months (comparing Fig. 1g–l with Fig. 1m–r). Thirty-four (of the 42) countries have higher average wind power availability during the nighttime than during the daytime. Solar power peaks in the middle of the day and drops off sharply to zero at dusk. The amplitude and duration of the daily cycles for solar power availability is consistently different during the summer and winter months across countries (Fig. 1g–l versus Fig. 1m–r). The daily cycle of solar resources is a barrier to realizing reliable solar-dominated electricity systems without energy storage and/or complementary wind generation to meet demand during the hours when the solar resource is not available. In addition, given our assumption of single-axis solar tracking, available solar power tends to be flat for several hours around its daytime peak during the daily cycles (Fig. 1g–r), though in some countries (e.g., Germany, South Africa, Australia) there is a consistent dip near noon, perhaps related to our adjustments of the direct radiation (details in Supplementary Note 1). Kendall's correlation coefficients of solar and wind resources in the 42 main countries range from −0.91 to −0.83 (see Supplementary Data 2),

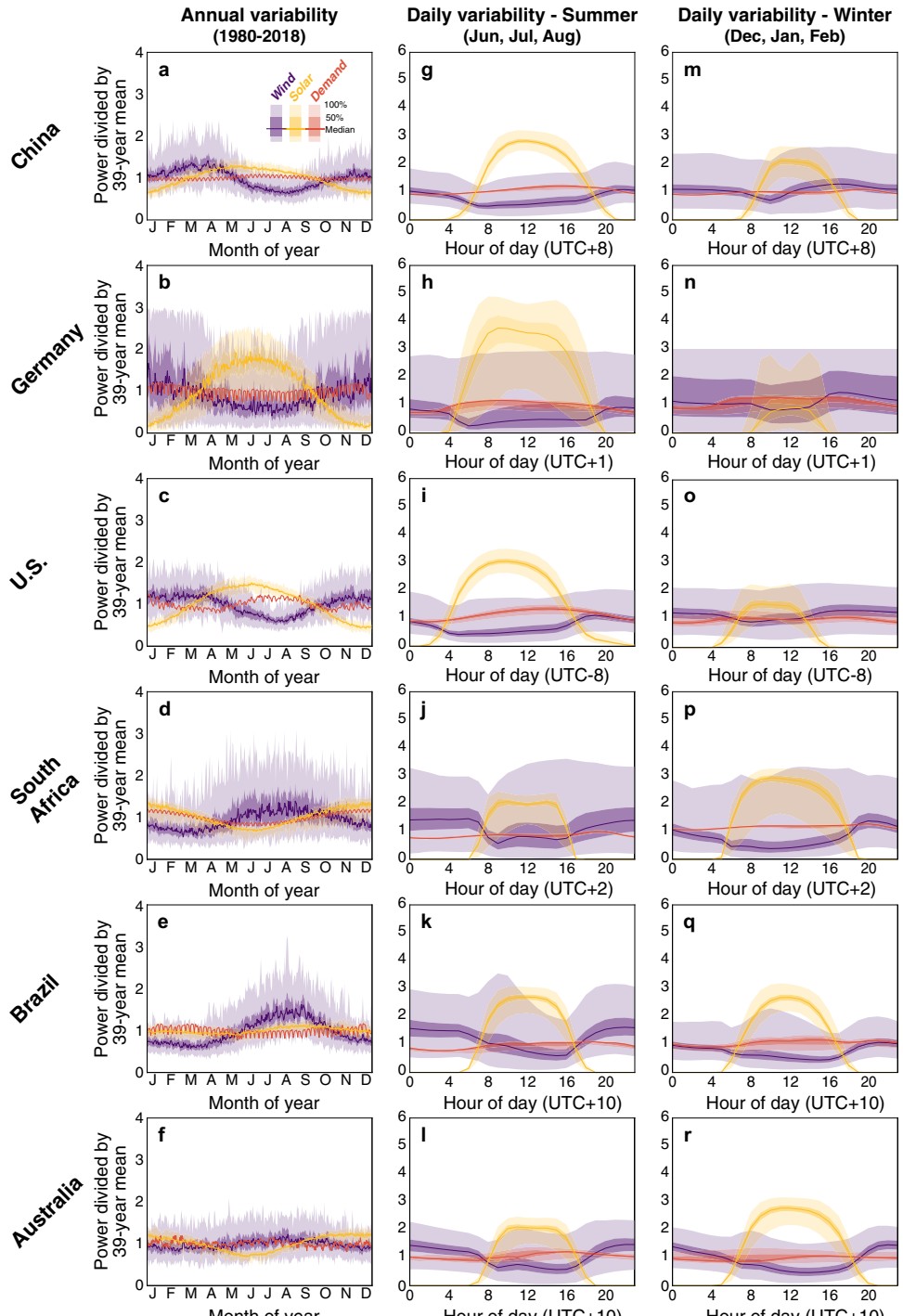

**Fig. 1 Temporal variability of solar and wind resources and electricity demand.** Climatological variability of the area-weighted median power from solar (orange) and wind (blue) resources for the selected country from six continents during the 39-year period 1980–2018. The countries (from the top row to the bottom row) are China (**a**, **g**, **m**), Germany (**b**, **h**, **n**), contiguous U.S. (**c**, **i**, **o**), South Africa (**d**, **j**, **p**), Brazil (**e**, **k**, **q**), and Australia (**f**, **l**, **r**). The left column (**a**–**f**) depicts the daily and seasonal variability, the middle column (**g**–**l**) depicts hourly summer (June, July, and August) variability, and the right column (**m**–**r**) depicts hourly winter (December, January, and February) variability. The lines represent the median, the dark shading represents the inner 50% of observations (25th to 75th percentile) and the light shading represents the outer 50% of observations (0th to 100th percentile) of the daily averaged value for that same day in each of the 39 years of record. Red curves in each panel represent electricity demand for a single, most recent, available year for each country. The time of day shown is the local time of each country and its relation to Coordinated Universal Time (UTC) is shown. Note that the middle of local time zones has been selected for the countries with multiple time zones. The solar, wind, and demand data are each normalized by dividing by their respective 39-year mean value.

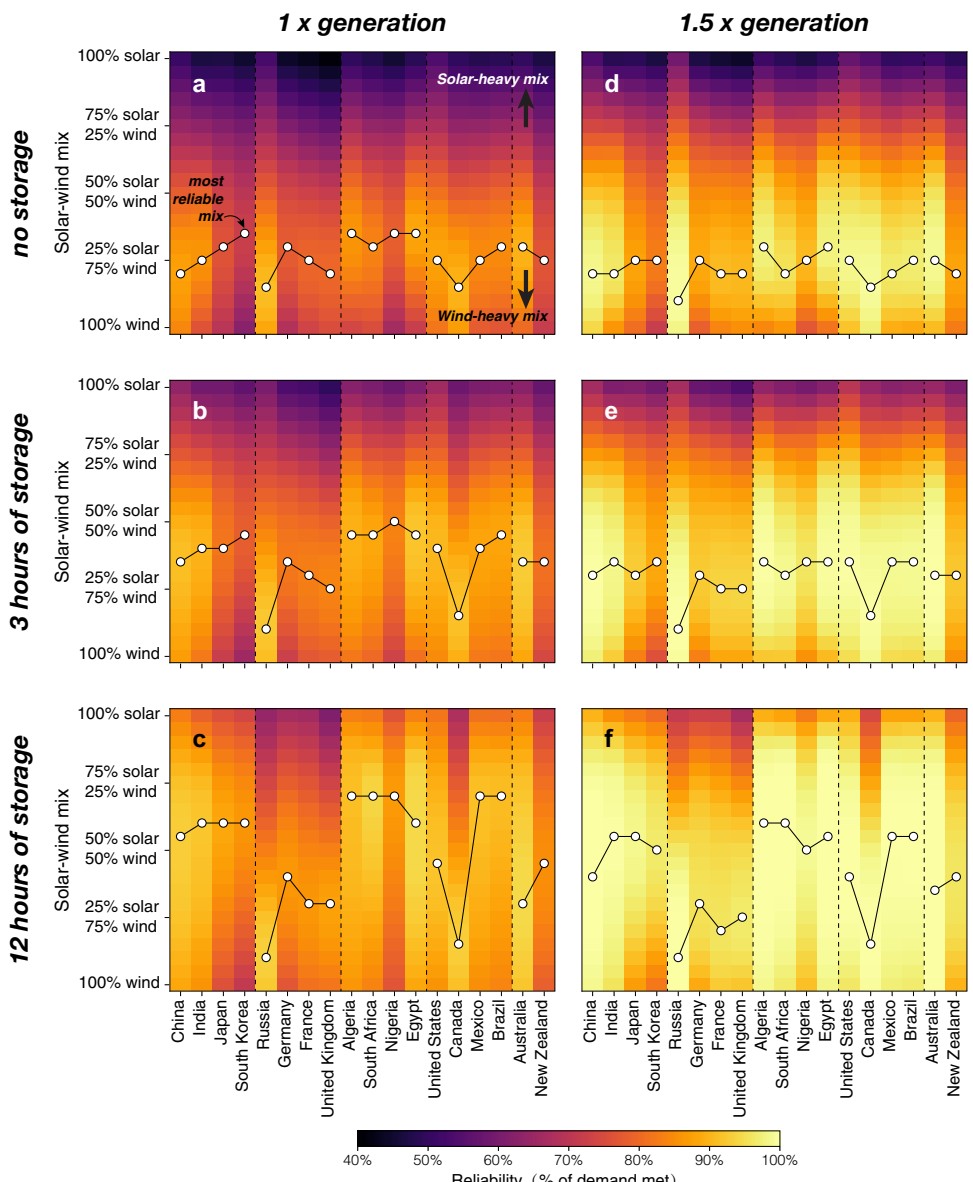

**Fig. 2 Reliability of electricity supply by varying the solar and wind resource mix, generation, and energy storage.** Shading in each panel represents the 39-year average estimated reliability (% of total annual electricity demand met) by a mix of solar and wind resources ranging from 100% solar to 100% wind (every 5% change for solar-wind generation mix). 18 main countries are chosen to show their ability to meet total annual electricity demand, including 16 main countries from four continents (Asia, Europe, Africa, and the Americas) and 2 main countries from Oceania. The white circles represent the highest reliability within each country under 21 sets of solar and wind generation mix (called the most reliable mix). Storage and generation quantities are varied in each panel: **a** 1x generation without storage; **b** 1x generation with 3 h of storage; **c** 1x generation with 12 h of storage; **d** 1.5x generation without storage; **e** 1.5x generation with 3 h of storage; and **f** 1.5x generation with 12 h of storage.

another indication of good complementarity (where −1 is the best possible complementarity)[20].

**The most reliable generation systems**. The colors in Fig. 2 show the reliability of electricity systems (i.e., the average percentage of electricity demand that is met each year from 1980 to 2018) based only on solar and wind resources for 18 major countries (4 from each of Asia, Europe, Africa, and the Americas, and 2 from Oceania; horizontal axes of each panel), according to: the mix of solar and wind generation (vertical axes), the level of annual generation relative to annual demand (1x in left panels and 1.5x in right panels), and the capacity of energy storage relative to mean electricity demand (0, 3, and 12 h in the first, second, and third rows of panels, respectively). Results for 24 other countries

are shown in Supplementary Fig. 3 and Supplementary Data 3. Figure 2a shows that without any excess annual generation or energy storage (assuming perfect national transmission), the most reliable mixes (white circles) of solar and wind generation could potentially meet 72–91% (average 83%) of electricity demand in these countries. Under these generation and storage assumptions, the most reliable solar-wind generation mixes range from 65 to 85% wind power (73% on average), with countries with substantial desert (like Algeria, Egypt, South Africa) favoring slightly more solar and less wind (65–70% wind) and with higher-latitude countries like Russia and Canada favoring more wind (85% wind; Fig. 2a).

Adding 3 h of energy storage, but still without excess annual generation, increases the reliability so that the most reliable mixes (white circles) meet 78–93% (average 87%) of electricity demand.

The share of solar generation in these most reliable mixes increases to 15–50% (36% on average; Fig. 2b). However, the share of solar generation increases less, or even decreases, in higher-latitude countries like Russia, Canada, and Germany (Fig. 2b). These trends continue as more storage is added, so that with 12 h of energy storage and no excess annual generation, 83–94% (average 90%) of electricity demand is met with mixes of 10–70% solar power (49% on average; Fig. 2c).

If generating capacities are instead increased so that annual generation exceeds annual demand in each country by 50% (i.e., 1.5x generation), but without energy storage, the most reliable mixes meet 83–99% (average 94%) of electricity demand. The 1.5x generation most reliable mixes are substantially more reliable than in the 1x generation systems but include more wind power: 70–90% wind power (78% on average; Fig. 2d). These "overbuilt" systems are more reliable in all of these 18 countries than the systems with 12 h of energy storage but no excess generation (Fig. 2c). Adding energy storage to systems whose generation is 1.5x annual demand

again increases both the system reliability (89–100%, average 98%) and the share of solar generation (most reliable mixes have 10–60% solar power, 36% on average; Fig. 2e, f).

**The unmet demand**. The scatter plots in Fig. 3 show the relationships among reliability, energy storage, excess annual generation, and countries' land area for the most reliable solar-wind mixes of all 42 countries analyzed (see relationships with a log y-axis in Supplementary Fig. 4). The linear fits in each panel show that solar-wind systems are generally less reliable in countries with smaller land areas (e.g., Fig. 3a). Specifically, our results across countries indicate that the reliability of solar-wind systems that lack energy storage increases by 7.2% for every factor of 10 increase in land area; this relationship further suggests the improvement in system reliability that might be expected by expanding transmission systems within large countries. However, excess annual generation tends to alleviate the disadvantage of

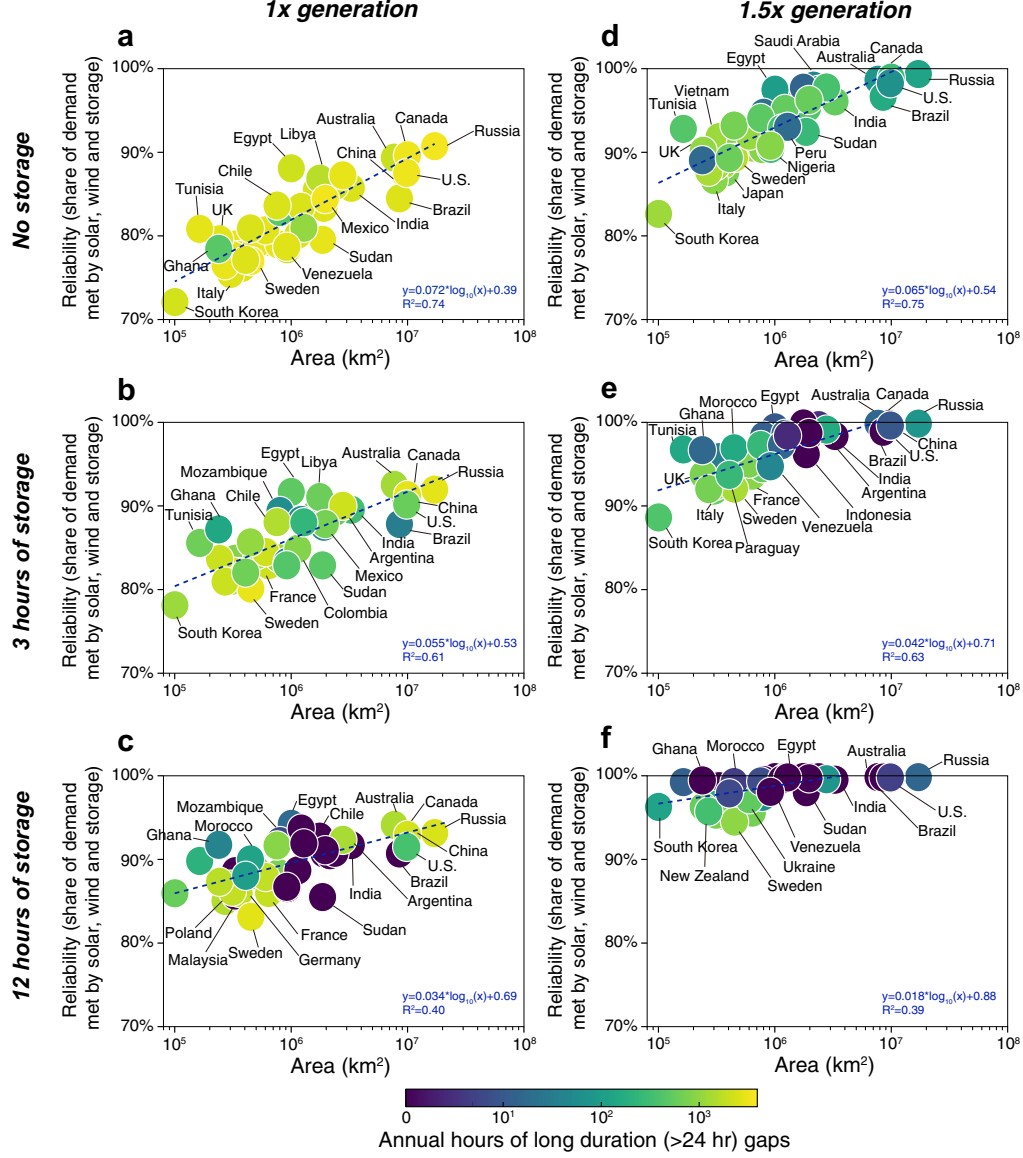

**Fig. 3 The relationship between the highest reliability of electricity supply system and country area among 42 major countries.** Shading of bubbles represents the annual average hours of long-duration (>24 h) power supply gaps. Storage and generation quantities are varied in each panel: **a** 1x generation without storage; **b** 1x generation with 3 h of storage; **c** 1x generation with 12 h of storage; **d** 1.5x generation without storage; **e** 1.5x generation with 3 h of storage; and **f** 1.5x generation with 12 h of storage.

small country area more than energy storage (this can be seen by comparing the slopes of the linear fits in panels of Fig. 3c and d). In addition, within each country, to compare the gains in reliability from excess annual generation and energy storage, a non-linear function was fit to the reliability given the land area, the level of annual generation, and the capacity of energy storage (see Supplementary Information). Our results indicate that a 10% increase in excess annual generation is equivalent to 3.9 h of storage (Supplementary Note 2).

Figure 3 also points to the nature of systems' unreliability: the color of bubbles indicates the average number of events in which there would be unmet demand in each of at least 24 contiguous hours (i.e., "long-duration gaps"). In systems that meet >95% of a countrie's demand, dozens of such long-duration gaps often remain each year (yellow and green circles). In some countries, excess annual generation reduces the number of such long-duration gaps more than adding 12 h of energy storage (e.g., compare Sweden, Australia, Canada, and Russia in Fig. 3c and d).

Figure 4 further characterizes the magnitude and duration of unmet demand in 16 major countries (removing two African countries from the 18 countries shown in Fig. 1 for figure symmetry; in descending order of their land area), with curves

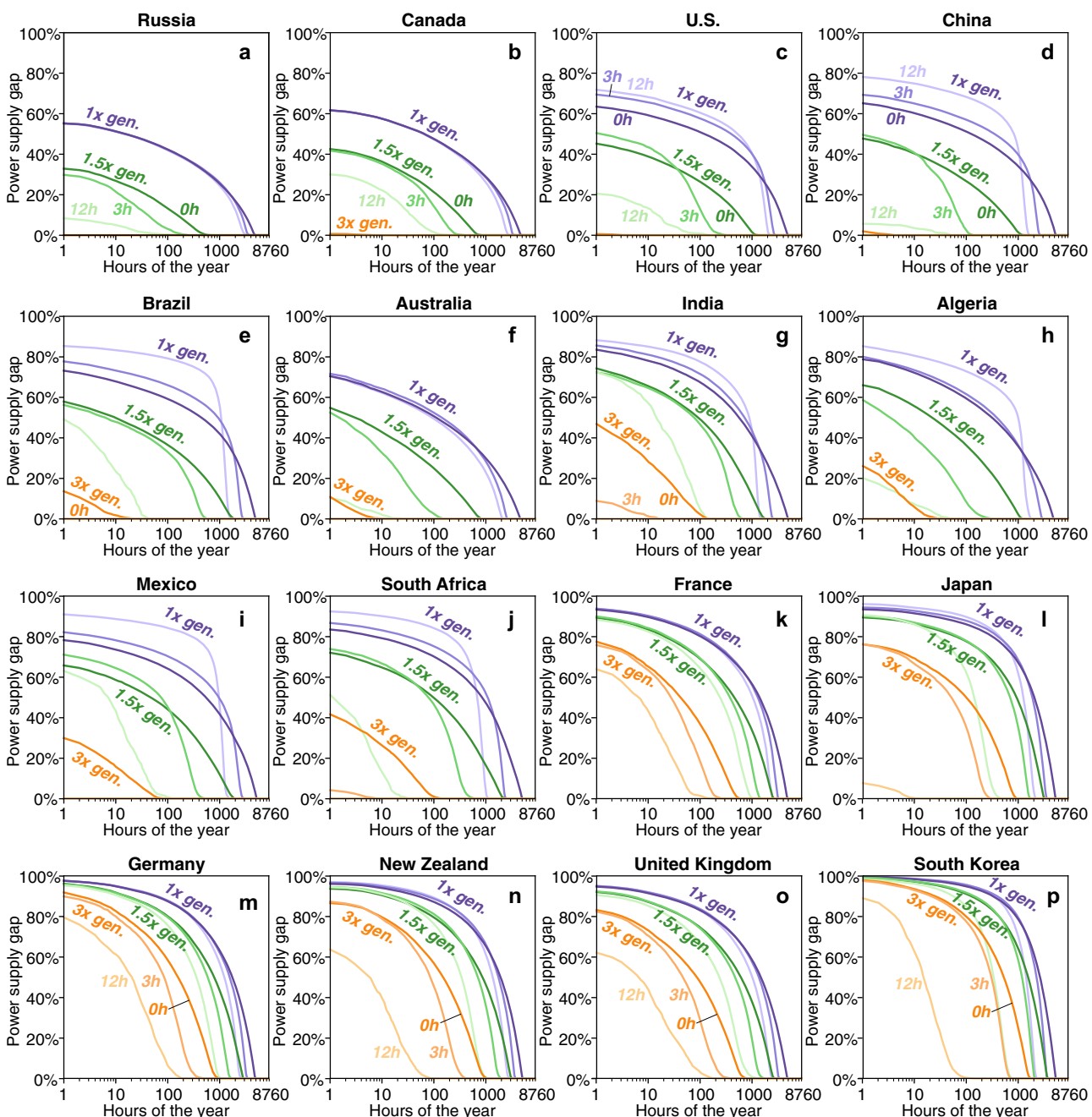

**Fig. 4 Average power supply gaps.** Areas under each curve show the share and hours of unmet electricity demand of the most reliable solar-wind systems in selected countries assuming specified storage and generation quantities: **a** Russia; **b** Canada; **c** contiguous U.S.; **d** China; **e** Brazil; **f** Australia; **g** India; **h** Algeria; **i** Mexico; **j** South Africa; **k** France; **l** Japan; **m** Germany; **n** New Zealand; **o** United Kingdom; **p** South Korea (see data in Supplementary Data 6). Color of lines represents different generation quantities: 1x generation in purple, 1.5x generation in green, and 3x generation in orange. Shading of lines represents different storage quantities: darkest shading represents without storage, medium shading represents 3 h of storage, and lightest shading represents 12 h of storage. Note that the y-axis of power supply gap represents the fraction of unmet demand to the total demand in that hour.

showing gaps of different system configurations sorted by their magnitude and according to the number of hours each year that such a gap occurred (power supply gap represents the fraction of unmet demand to the total demand in that hour averaging over 1980–2018; see relationships with a log y-axis in Supplementary Fig. 5). For example, the pale purple curves show that systems with no excess annual generation and 12 h of energy storage consistently have gaps in which >50% of demand is unmet for >1000 h per year (Fig. 4). Pale green curves show that systems with 50% excess annual generation and 12 h of energy storage may have much smaller and shorter gaps in some countries (e.g., <10% of demand unmet in fewer than 100 h per year in Russia, China, and Australia), but the gaps may still be >20% of demand for tens of hours or more in countries with relatively large land areas (e.g., Canada, Brazil, India, and Mexico) and >60% of demand for several hundred hours per year in countries with smaller areas (e.g., France, Japan, Germany, New Zealand, the U.K., and South Korea; Fig. 4). Indeed, in smaller countries, substantial gaps (>30% of demand for >20 h per year; pale orange curves in Fig. 4) remain in systems even with 12 h of energy storage and annual generation that is 3x annual demand.

**Benefits from sharing resources of multiple nations**. We also evaluate the reliability benefits of regional electricity interconnections whereby the solar and wind resources of multiple nations are pooled and shared, again assuming perfect transmission within these regions. The maps in Fig. 5 present the effects of such spatial aggregation, showing the highest reliability of solar-wind generation with no excess annual generation or energy storage at the national level (Supplementary Data 7; Fig. 5a), as well as when a system is aggregated into 19 separate, contiguous multinational regions (Fig. 5b; categorization in Supplementary Data 4) and 6 continents (Supplementary Data 7; Fig. 5c). Each step produces substantial improvements in reliability, with >89.8% of hourly demand met everywhere when resources are aggregated at the continental level (Fig. 5c). Figure 5c also indicates the additional reliability gains in these systems that would be achieved as a consequence of specific intercontinental connections. Supplementary Fig. 6 shows that the supply gaps in continental-scale solar-wind systems might be entirely eliminated in Africa, Asia, and South America, and limited to <2% of demand and 49, 26, and 13 h in Europe, Oceania, and North America, respectively, given excess annual generation of 50% and 12 h of storage. Substantial supply gaps

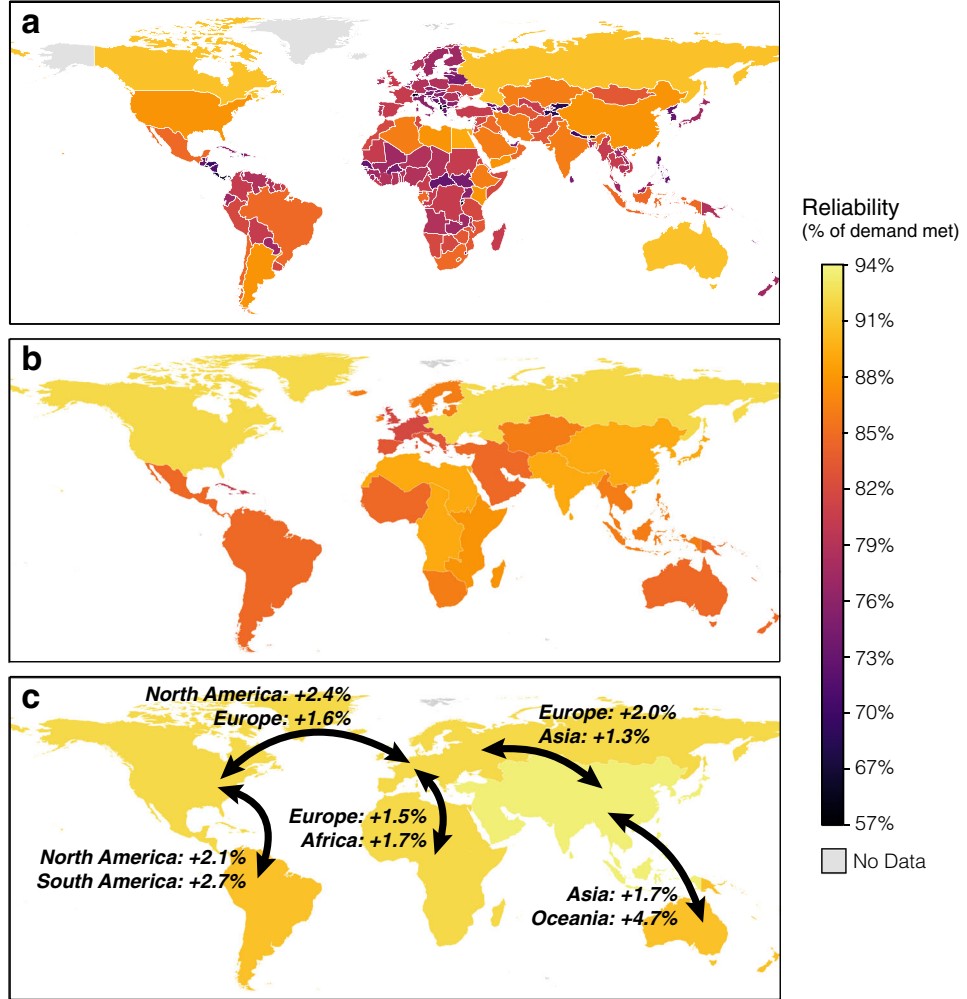

**Fig. 5 Maps of electricity system reliabilities under the most reliable solar-wind mix without excess generation or energy storage.** Maps show the reliability (i.e., hourly averaged resource adequacy) at country/region scale (**a**; Supplementary Data 4), the subcontinent scale (**b**; 19 multinational regions, and listed in the SI), and at continental scale (**c**; 6 continents: Asia, Europe, Africa, North America, South America, and Oceania). We also evaluated the reliability of the power supply system assuming several intercontinental connections (shown as the arrows: Asia–Oceania, Europe–Asia, Europe–Africa, North America–Europe, and North America–South America). The added reliabilities for each continental power system under various connections are labeled.

remain for continental-scale systems when excess annual generation and energy storage are not available (Supplementary Fig. 7).

## Discussion

Our results suggest that, neglecting transmission constraints, with systems sized to meet time-integrated annual electricity demand, major countries' solar and wind resources could meet at least 72% of instantaneous electricity demand without excess annual generation or energy storage. For instance, in the contiguous U.S., a solar and wind power system could provide ~85% of total electricity demand, which is consistent with the prior studies and reports[12,25]. Solar and wind resources can achieve greater levels of reliability by adding energy storage, increasing deployed capacities (i.e., generating electricity in excess of annual demand), or pooling resources of contiguous, multinational regions[26]. However, the marginal improvements in reliability related to these options differ considerably across countries and regions, according to their land area, location, and geophysical resources (Supplementary Figs. 8 and 9).

In small, high-latitude countries, the highest reliability systems are usually wind-heavy (e.g., as high as 95% wind power), with particularly large reliability gains achievable by regional aggregation. In contrast, the most reliable systems in temperate/tropical countries tend to include more solar. Meanwhile, the most reliable systems are not always the same systems that would minimize the frequency of long-duration (≥24 h) power supply gaps (Supplementary Fig. 9). In general, more solar in the wind-solar mix reduces the frequency of long-duration gaps. Although reasonably high levels of reliability can be reached by solar-wind resources alone, the defining challenge of such systems are the longer-duration gaps, often associated with extreme weather episodes. For instance, historical solar and wind resources data in Germany reveal that there were nearly 2 weeks in which dispatchable generation had to cover practically all of the demand because of a period with very low solar and wind power availability (called "dark doldrums")[27]. Although with vast enough wind and solar capacity it might still be possible to meet demand in all hours, the required capacity increases exponentially after a point that depends on the renewable resources of that country, and it is this geophysically-dependent point that will largely determine the cost-effectiveness of highly-reliable, renewables-based electricity systems. Although dispatchable fossil fuel generators with 100% effective carbon capture storage (CCS) could provide system reliability without emissions[2], such underutilized and capital-intensive backup electricity would require higher investments and variable costs. In contrast, combustion turbines or combined cycle plants burning carbon-neutral biogas, syngas, or hydrogen might have comparatively low capital costs, but would require additional and large capital investments to produce such fuels (e.g., biodigestion, direct air capture, Fischer-Tropsch, and/or electrolysis). Sector-coupling or right-sizing of these net-zero emissions fuel-production facilities could nonetheless make infrequent operation of generators feasible[28]. More firm generation would mean less solar and wind capacity in a given system, which might or might not be cost-effective depending on technology costs. But many jurisdictions and advocates are interested in "maxing out" solar and wind. Our results are especially relevant in that context, highlighting the implications of country-level differences in the variability of solar and wind resources, including how much storage and firm generation might be required to ensure resource adequacy. Although our methods are simple and transparent, our goals and findings are remarkably consistent with much more complex approaches. For example, the recently published Net-Zero America report includes a cost-

optimized "all-renewables" scenario which decarbonizes U.S. electricity without nuclear or CCS: by 2050, ~81.6% of primary energy in the E + RE + scenario is from solar and wind[29].

Our analysis has important limitations and uncertainties. To improve the generality of our results, our analysis focuses exclusively on geophysical constraints and does not consider economic feasibility. As noted throughout, our reliability estimates are a best case given the assumption that electricity can be transmitted losslessly throughout a region of interest. Also, we use area-weighted averages for solar and wind generation potential without regard to existing protections or uses. This use of area-weighted averages affects our estimates in two important ways. First, our estimates may include areas where currently generation cannot be sited. Second, our derivation of solar and wind capacity factors implies uniform distribution of wind and solar generation technology (i.e., a horizontal single-axis tracking system applied in this work), which does not allow us to select locations with particularly high capacity factors or to strategically select a set of locations whose generating potential is mutually negatively correlated. This second point has the effect of making our estimates for the efficacy of solar and wind resources to meet electricity demand more conservative by using the horizontal single-axis tracking system compared to the dual-axis solar tracking systems. For this case, dual-axis solar tracking systems are added to test the impacts on the system reliabilities (see Supplementary Note 3), we find that different solar tracking systems have very small impacts on the electricity system reliabilities and the reliability change ratios are within ±5% under the 1x generation system and less sensitive under 3x generation system (Supplementary Fig. 10). However, either method to calculate capacity factors of national and regional area-weighted averages may also reduce the resource variability and thereby increase estimates of reliability. Third, hourly variations of solar and wind capacity factors in the reanalysis data MERRA-2 we used may be biased. A new analysis based on a different and independent reanalysis product, ERA5[30,31], has been added and compared to the original results (see Supplementary Note 3 and Supplementary Figs. 11-12). Our estimates of the system reliabilities by using ERA5 data in the 42 major countries are in good agreement with results of MERRA-2: under 1x generation and the most reliable mixes without storage, reliability under the different loads varies on average from −9.4 to 1.3% (see Supplementary Fig. 9a). The differences are similar in systems with excess generation (Figs. S11b-c). We also compared the magnitude and duration of unmet demand in 16 major countries like Fig. 4 (see Supplementary Fig. 12). The data products of MERRA-2 and ERA5 both can essentially capture the number of hours each year that such a gap occurred. By contrast, the MERRA-2 data has a better performance of meeting hourly demand in larger countries (i.e., Russia and Canada) but a similar performance in small countries (i.e., United Kingdom). The somewhat different patterns of resource variability in the two datasets do not alter our main conclusions.

Our estimates show that the marginal reliability benefit of increased capacity of storage or increased overbuild of wind/solar declines steadily. Under a given capacity of energy storage (e.g., 3 h), our results of 1x, 1.5x, and 3x generation show that the first 10% excess generation increase is larger than the next 10% excess generation increase (i.e., the marginal benefit for system reliability decreases as excess generation increases). As might be expected, the diminishing marginal benefits between excess generation and increased storage apply in both directions. Our fitting model performs well across different nations, overbuild levels, and storage levels. The differences in reliability between the estimates and the model predicted values are between −5.5 and 5.8% and ~80% of the differences are within ±2%, with no systematic bias related to region or the magnitude of overbuild or storage.

Nonetheless, our model and conclusions are limited by our experimental design and the discrete levels of excess generation (1x, 1.5x, and 3x) and storage (0, 3, and 12 h) we evaluated.

We compare the reliability improvements obtainable by energy storage, excess capacity, and regional aggregation but not the relative costs of the different options. For example, the energy storage capacities we consider are in some cases quite large: energy storage equal to 12 h of mean electricity demand in the contiguous U.S., Germany, and Japan represents 5.6, 0.7, and 1.4 TWh, respectively (Supplementary Data 5). These combined storage capacities represent ~35 times the capacity of Li-ion batteries produced globally to date[32], and more than 200 times the pumped hydro storage capacities that now exist in those countries[33]. The feasibility of 12 or more hours of energy storage may depend on continued innovation and learning related to the associated materials and technologies[34–37]. Similarly, the feasibility of pooling solar and wind resources over national or multinational regions may depend on both technological advances that reduce the costs, losses, and risks of power transmission[38–40] as well as shifts in the socio-political support for such infrastructure[41,42]. In addition, setting up purely solar and wind supplied electricity systems requires a large number of solar panels and wind turbines to be installed, and we do not incorporate the impacts or interactions (e.g., wakes) from these hypothetical build-outs. Last, in this work, only 1-year of demand data is employed to assess the geophysical constraints of 39-year solar and wind resources. On one hand, we understand that the hourly patterns of countries electricity demand will of course change over time with changes in population, economic activities, power generation structure, and technology. For example, stronger positive correlation between solar/wind availability and demand may be observed as renewable energy gradually dominates the power system. However, our analysis compares resources and demand in different years and at the country-level, which should preclude any bias related to specific subnational weather events. On the other hand, electricity demand profile may also dramatically change with future high electrification. We therefore perform additional analysis using the demand pattern from the future high electrification scenario (i.e., combining the high electrification scenario and rapid technology advancement)[43] and use the results to discuss the sensitivity of our results to such different load profiles (see Supplementary Fig. 13), and the results of this test for the U.S. show that reliability is not especially sensitive to the high electrification demand profile: under 1x generation without storage, reliability under the different renewable mixes varies on average from −1 to 2.5%. The differences are even smaller in solar-heavy systems with excess generation. In addition, we test the sensitivity of our results to such changes in demand by simulating the reliability of U.S. resources in meeting current loads from each other region (see Supplementary Note 4). These tests show that reliability is not especially sensitive to demand profiles: under 1x generation and the most reliable mixes without storage, reliability under the different loads varies on average from −9 to 2% (see Supplementary Fig. 14a). The differences become even smaller in systems with excess generation (Supplementary Fig. 14).

Despite these simplifying assumptions, our results offer insights from those provided by multi-year time step integrated assessment modes (IAMs) or hourly, cost-optimized energy system models. Specifically, hourly resolution over several decades allows us to evaluate the adequacy of regional solar and wind resources independent of costs. For example, cost-optimizing models which either require renewables sources to meet a very high share of demand or else assume extremely cheap renewable costs generally find substantial increases in system costs related to, e.g., energy storage. Our geophysically-focused results help to explain such results irrespective of cost assumptions. Indeed, we compare the estimates of reliability and capacities in this study with several techno-economic studies that have used independent approaches to model regional solar- and wind-dominated electricity systems in detail[29,44,45]. In each case, focusing on the U.S., these studies find that the share of non-emitting (or carbon neutral) electricity contributed by solar and wind in cost-optimized systems is typically ~80%, with the residual demand for non-emitting generation met by firmer renewables such as biomass, hydroelectricity, and geothermal[29,44,45].

Variable solar and wind energy are projected by many to be the dominant sources of electricity in net-zero emissions energy systems of the future. With solar and wind capacities sized such that total annual generation meets total annual demand, seasonal and daily complementarities of these resources make them capable of meeting three-quarters of hourly electricity demand in larger countries. Increasing the share of demand that can be met by solar and wind generation will require either "overbuilding" (i.e., excess annual generation), the introduction of large-scale energy storage, and/or aggregating resources across multinational regions (Supplementary Data 6). We highlight the geophysical considerations related to these options, but economics and geopolitics will also strongly influence which strategies are ultimately adopted and are therefore important topics for further research. Our analysis for the 39-year record of solar and wind resources is in part to obtain a statistically significant analysis of interannual variability and rare events (such as prolonged storms). Establishing estimates for interannual variability and the frequency of rare events that impact solar and wind generation potential is important when considering the lifetime of the capital asset stock in an electricity grid and requires the use of many years of data. Our normalized analysis of the reliability for purely solar and wind supplied electricity system would apply as well to a system with other slowly time-varying generation (e.g., coal, hydro, geothermal, or nuclear) because the variability of solar and wind generation and related long-duration gaps in electricity supply will have to be managed either by ramping backup technologies up and down or by curtailing excess solar and wind generation. Our results reveal national and regional differences in solar and wind resources that may be useful to decision makers and researchers prioritizing their investments in pursuit of reliable and cost-effective electricity systems based predominantly on solar and wind energy.

## Methods

**Hourly solar and wind capacity factors**. The capacity factor describes the actual energy output as compared to the systems' rated energy output (power capacity multiplied by 1 h)[12]. To calculate the wind and solar capacity factors for this study, we first obtained the hourly climatology data from the Modern-Era Retrospective analysis for Research and Application, Version-2 (MERRA-2) reanalysis product, which spans 39 years (1980–2018) and has a horizontal resolution of 0.5° by latitude [−90–90°] and 0.625° by longitude [−180–179.375°] with 361 × 576 grid cells worldwide[21,22]. Here we used the surface incoming shortwave flux [W m$^{-2}$] (variable name: SWGDN), top-of-atmosphere incoming shortwave flux [W m$^{-2}$] (variable name: SWTDN), and surface air temperature [K] (variable name: T) for deriving solar capacity factors; and wind speed at 100 m [m s$^{-1}$], estimated based on wind speed at 10 m and 50 m (variable names: U10M, V10M, U50M, and V50M) and a power-law relationship, to derive wind capacity factors. Wind and solar capacity factors were calculated for each grid cell and each hour. Each raw data point (an hourly energy density (solar) or wind speed (wind) value at a specific location and time) was then converted into the corresponding capacity factor based on the following procedures.

For solar capacity factor, we first calculated the solar zenith angle and the solar incidence angle for each grid based on the latitude/longitude location and local time[46,47], and then estimated the in-panel solar radiation[48]. Here we separated the direct and diffuse solar radiation components based on an empirical piecewise model[49] that takes into account both ratios of surface to top-of-atmosphere solar radiation (i.e., the clearness index) and the local zenith angle. We assumed a horizontal single-axis tracking system (north-south direction) with a tilt of the solar panel to be 0° and a maximum tuning angle of 45°. Solar power output from a

given panel is calculated using the performance model described by Huld et al.[50] and Pfenninger and Staffell[51], which considers both the surrounding temperature and the effect of irradiance. It is noted that we assumed the single-axis trackers for calculating solar capacity factors, which may be unsuitable enough for the small countries such as Japan, South Korea, and United Kingdom. These small countries do not have enough uncommitted land area for that and are going very likely to have to favor no tracking with rooftop photovoltaic system. Therefore, we further assessed the impacts of different solar tracking systems (i.e., single-axis and dual-axis for both a horizontal and a vertical axis) on the electricity system reliability. The detailed comparisons are shown in the SI (Supplementary Note 3 and Supplementary Fig. 10).

For wind capacity factor, by assuming a wind turbine hub height of 100 m, the raw wind speed data is first interpolated to 100 m by employing a power law, based on wind speed at 10 and 50 m. The 100-m-height wind speed is estimated by employing the following Eqs. (1) and (2):

$$\alpha = \frac{\log(U_{50,i}) - \log(U_{10,i})}{\log(50) - \log(10)} \tag{1}$$

$$U_{100,i} = U_{10,i} * \left(\frac{100}{10}\right)^{\alpha} \tag{2}$$

where $i, \alpha$ represent grid and alpha exponent for wind profile, and $U_{10}, U_{50}$, and $U_{100}$ represent wind speed at 10, 50, and 100 m.

The wind capacity factor calculation employed a piecewise function consisting of four parts: (i) below a cut-in speed ($u_{ci}$) of $3\,\mathrm{m\,s^{-1}}$ the capacity factor is zero, (ii) between the cut-in speed of $3\,\mathrm{m\,s^{-1}}$ and rated speed ($u_r$) of $12\,\mathrm{m\,s^{-1}}$ the capacity factor is $u_{ci}{}^3/u_r{}^3$, (iii) between the rated speed of $12\,\mathrm{m\,s^{-1}}$ and the cut-out speed ($u_{co}$) of $25\,\mathrm{m\,s^{-1}}$ the capacity factor is 1.0, and (iv) above the cut-out speed of $25\,\mathrm{m\,s^{-1}}$ the capacity factor is zero[12,52,53]. The process yielded the solar and wind capacity factors for each grid cell and each hour.

An area-weighed mean hourly energy generation profile was created for the solar and wind resources individually for each region of interest. In this work, hourly solar and wind capacity factors for 168 regions/countries were produced. Capacity factors derived from reanalysis data were known to differ from real-world systems[12,54], and thus these calculated capacity factors from the reanalysis dataset were rescaled. That is, the reanalysis data were used herein only for reflecting the temporal and spatial characteristics of the resource. For consistency, we normalized capacity factor values using the 25th percentile calculated capacity factor data for a region of interest due to data availability of real-world wind and solar capacity factors from public datasets or reports for all the countries and regions of interest. Our estimates represent real-world wind and solar capacity factors that are in good agreement with available observational data[55]. We then obtained the time-series hourly normalized wind and solar capacity factor dataset at the country/region level.

**Country-level hourly electricity demand data**. In this work, country-level hourly electricity demand data were estimated in various ways, such as from government and electricity market websites, public power systems datasets, and previous studies (Supplementary Data 1). As shown in Supplementary Data 1, we compiled 168 countries and regions' demand data, including real-word hourly demand data of 62 countries and regions, and projected hourly demand data of the rest due to data availability[56]. Toktarova et al. developed a multiple linear regression model to project electricity demand in hourly resolution for all countries globally by incorporating 57 real load data profiles of diverse countries to analyze the cyclical pattern of the data. In addition, given the different self-consistent continuous gapless time series of hourly electricity demand among different countries and regions, a single latest year of hourly electricity demand data was used in our following simulations to investigate the impact of diversity of solar and wind resources across years on power system reliability for countries and regions with available real-world electricity data. For the rest of the countries and regions, we chose the hourly electricity demand data of the most recent year of future (i.e., the year of 2020) from the projection model[56]. We herein obtained the country-level demand dataset by joining the 1-year hourly demand data together 39 times to form a 39-year record consistent with the resource data. In addition, for regional demand data, we combined all the demand data of available countries within the according region at hourly scale to represent the temporal characteristics.

**Simulation design**. A set of forward simulations were performed to track the ability of wind, solar installed capacity, and energy storage, if present, to meet demand in every hour. In this study, we used a Macro Energy Model (MEM), which is developed for optimizing electricity system (or electricity and fuels) without considering any spatial variation, policy, capacity markets[57]. Without considering any power system cost, generation technology, and transmission loss, we modeled the idealized hourly power supply process through dispatching wind and solar energy, as well as charging or discharging of storage, if present. Here we specified the wind and solar installed capacity, and storage capacity under the various capacity mixes of solar and wind fractions (i.e., every 5% change of solar fraction from 0% solar and 100% wind to 100% solar and 0% wind) and different levels of excess annual generation (i.e., 1x, 1.5x, and 3x generation) and energy storage (i.e., maximun 3 and 12 h of charging time) assumptions. The installed

capacities for solar, wind, and storage for individual countries/regions are estimated using the Eqs. (3)–(5).

$$Capacity_{solar,y} = \mathrm{SF} \times \mathrm{OB} \times \mathrm{Pwr\_avg}_y \times \mathrm{Hrs}_y \Big/ \sum_y \mathrm{CF}_{solar} \tag{3}$$

$$Capacity_{wind,y} = (1 - \mathrm{SF}) \times \mathrm{OB} \times \mathrm{Pwr\_avg}_y \times \mathrm{Hrs}_y \Big/ \sum_y \mathrm{CF}_{wind} \tag{4}$$

$$Capacity_{storage,y} = \mathrm{Pwr\_avg}_y \times \mathrm{Bat}_s \tag{5}$$

where $y$ and $s$ represent the year and size, respectively. $Capacity_{solar}$, $Capacity_{wind}$, and $Capacity_{storage}$ represent the solar, wind, and storage capacities, respectively. SF represents the fraction of energy generated from solar (from 0 to 100% at intervals of 5%); OB represents the overbuilding of capacity, equaling 1, 1.5, or 3; Pwr_avg represents the mean power demand; Hrs represents the total hours in the year; $\mathrm{CF}_{solar}$ and $\mathrm{CF}_{wind}$ represent normalized capacity factors of solar and wind, respectively. And Bat represents battery storage, equaling 0 (i.e., no storage), 3, or 12.

When storage was assumed to be available, we assumed the initial status of storage was the same as the final status for each year, which means the charging and discharging process is balanced. We also assumed a storage charging round-trip efficiency and storage decay rate of, respectively, 90% and $1.14 \times 10^{-6}$ per hour (i.e., 1% of stored electricity lost per month)[2], reflecting the high-end performance of current batteries[58,59]. Dispatchable energy used to charge a battery (called the maximum hourly storage charging) was no more than the storage power rating, equaling storage capacity divided by storage charging time.

Given the restriction of computing resources, we chose ten major countries by comprehensively considering the electricity demand and growth domestic product (GDP) from each continent except Oceania (i.e., Asia, Africa, Europe, and the America), within which only two main countries were selected (i.e., Australia and New Zealand). For each main country, 21 sets of the solar and wind mix from 0% solar and 100% wind to 100% solar and 0% wind with 5% change under 3 groups of overbuilt (1x, 1.5x, and 3x generation) and 3 groups of storage (no storage, 3 h, and 12 h of storage) were simulated, totaling 7938 simulations for all the main countries. To investigate the ability to supply power at multinational regions, continental, and intercontinental scales, we further applied the same simulation design for the main countries to multinational regions, continents, and multi-continental regions (Supplementary Data 4). In addition, except the abovementioned main countries, 103,194 one-year simulations consisting of 21 sets of the solar and wind mix with no excess generation or energy storage, were added for each of the remaining 126 countries worldwide.

**Hourly electricity supply process**. For only solar-wind electricity systems without storage, in a given hour, the MEM model estimates the ability of power to be produced by assessing whether dispatchable solar and wind energy is no less than electricity demand. Excess solar and wind energy can be curtailed due to no available storage. 100% reliability results if the solar and wind power supply system can meet all the electricity demand in every hour of the simulation.

When storage was assumed to be available in a given hour, if the solar and wind energy could meet the electricity demand, storage would be charged with excess solar and wind generation, if available, until the storage is full under the constraint of the maximum hourly storage charging, after which solar and wind energy can be curtailed. In contrast, if wind and solar energy cannot meet electricity demand, storage would be discharged to fill the power supply gap until storage is emptied or the power supply gap is filled.

Here, we define reliability assuming electricity systems use only wind/solar/storage resources to meet current demand for electricity services. If one allows for other backup electricity (e.g., using natural gas with or without CCS), then issues of reliability with excess annual generation and/or storage are largely moot.

## Data availability
The electricity demand, solar, and wind capacity factors data generated for this study have been deposited in Dantong2021/Geophysical_constraints: Data of electricity demand, solar and wind capacity factors (v1.0). Zenodo. https://doi.org/10.5281/zenodo.5463202.

## Code availability
The Macro Electricity Model (MEM) code is available on GitHub via https://github.com/ClabEnergyProject/MEM.

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

## Acknowledgements

This work was supported by the National Natural Science Foundation of China (41921005). D.T., D.J.F., L.D., and K.C. were supported by the Carnegie Institution for Science endowment and a gift from Gates Ventures, Inc. S.J.D. was supported by the US

National Science Foundation (Innovations at the Nexus of Food, Energy and Water Systems (INFEWS) grant EAR 1639318).

## Author contributions

K.C., N.S.L., S.J.D., and D.T. designed the study. D.T. performed the analyses, with support from D.J.F. on simulations and L.D. on resources estimates, and from S.J.D., Q.Z., N.S.L., and K.C. on analytical approaches. D.T. and S.J.D. led the writing with input from all coauthors.

## Competing interests

The authors declare no competing interests.
