## [Peer Review File · Nature Communications]

Geophysical constraints on the reliability of solar and wind power worldwideREVIEWER COMMENTS

Reviewer #1 (Remarks to the Author):

Like the earlier work, Shaner et al., this is an interesting and potentially insightful contribution to the literature and I would welcome publication. However, I have three main/important methodological (or perhaps philosophical) questions about the work.

1. Most importantly, are you really exploring/revealing any truly any fundamental geophysical constraints here? Isn't it a matter of economics in the end that will constraint (or land area/siting challenges that may bind even before economics)? So long as there are no hours with both zero solar and zero wind output, it should be mathematically possible to create a convex combination of wind and solar capacities that meet demand in all hours. If you add storage to the mix, then this gets easier. Even in countries with one or more hours of zero combined wind and solar output (e.g. a becalmed night), storage capacity can shift output from other periods to this time and achieve supply/demand balance. So what is the geophysical constraint or limit? There isn't really one, is there? You are revealing geophysical *considerations* or *dynamics* inherent in the temporal variability of wind and solar at different locations and spatial scales and across interannual variability. This is insightful in and of itself, but the real limits or constraints on renewables-dominated systems come from the economic and/or socio-political limitations that arise, in part, from these geophysical dynamics. (I had the same thoughts upon reading Shaner et al., and this new work raises them all again).

2. How do you validate your time series data (wind and solar simulations) and correct for biases in the hourly profiles of wind (and solar if relevant) in the reanalysis data? Reanalysis data series are known to have some challenges capturing in particular diurnal patterns of wind power output in some regions. The methods discussion mentions trying to correct annual capacity factors, which is a good step. But what about temporal patterns? As this is your focus here (hourly resolution variability in wind and solar series), this seems key, not just adjustments to correct bias in annual capacity factors.

3. I would expect that the marginal reliability benefit of increased capacity of storage or increased overbuild of wind/solar declines steadily (and perhaps rapidly, based on earlier works on these topics). You do not explore these marginal dynamics at all in your work, which I would think would be one of the main benefits of taking a simplified approach as in this study (without cost optimization): you can trace out marginal changes across varying levels of overbuild and/or storage capacity (perhaps two-d or isoquant plots showing both dimensions). You present some of these factors as if they are fixed/constant, but that is unlikely. For example, p. 5 states "Our results indicate that a 10% increase in excess annual generation is equivalent to 3.9 hours of storage" but is this substitution ratio consistent? Does it vary across nations and at different overbuild levels? I would expect this to change on the margin, so first 10% excess generation might be worth 3.9 hours, but next 10% would be worth less, etc. There should be diminishing marginal returns here.

-Signed,
Jesse D. Jenkins

These are my major/substantive concerns with the work.

Specific comments follow:

p. 1, abstract: "If future net-zero emissions energy systems rely heavily on solar and wind resources, spatial and temporal mismatches between resource availability and electricity demand may degrade system reliability." Add "or require installation of sufficient firm generation capacity and/or long-duration energy storage."

p. 1, "in most scenarios" - revise to "most cost-optimized scenarios"?

p. 1 "The efficacy of meeting electricity demands with generation from solar and wind resources

depends on factors such as location and weather; the area over which generating assets are distributed; the mix and magnitude of solar and wind generation capacities; and the availability of energy storage" - insert "and firm generation capacity." Consider adding cite to <https://doi.org/10.1016/j.joule.2018.08.006> or similar work on firm generation.

p. 1, "Resource adequacy planning standards are also high: in North America (BAL- 502-RF-03)17, generating resources must be adequate to provide no more than 1 day of unmet electricity demand in 10 years (i.e. 99.97%)." Note that different jurisdictions define 1 in 10 standard differently. Some define it as one day in 10 years (so 99.97%) but some define it as one loss of load event (of unspecified duration, as short as one hour) per 10 years, which implies generally higher reliability (closer to '4 nines' or 99.99%). See <https://pubs.naruc.org/pub/FA865D94-FA0B-F4BA-67B3-436C4216F135#:~:text=As%20recognized%20in%20the%20recent,reliability%20standard%20has%20different%20interpretations.&text=one%20event%20in%2010%20years,LOLE%20in%20events%20per%20year>.

p. 1, "Our goal is to identify the opportunities and challenges of variable renewable resources in greater detail than can be done by integrated assessment models that have multi- year time steps." How is this work situated relative to cost-optimized capacity expansion planning models with hourly chronological operating detail (e.g. Switch, Pypsa, GenX, urbs, OpenCEM, etc)? And why is your approach the appropriate one for this purpose? I'd also suggest expanding this first comparison point to both IAMs and energy-economic models like NEMS or TIMES/MARKAL, as your are equally valid in your comparison to both here.

p. 2, "the great electricity demand" - should say "greatest" not "great"

p. 2, "Electricity demand profiles for each country are determined by factors such as economic conditions, prevailing weather conditions and consumer usage patterns" - Note for the reader that you are using historic demand patterns and are not capturing plausible changes in demand due to electrification, managed charging/flexible loads, or introduction of electrolysis or other large flexible demands in future years concurrent with high penetrations of renewables. Another important limitation to note. Electrification can dramatically change demand profiles, particularly in northern latitude countries not currently dependent on electric heat. This changes both the diurnal and seasonal patterns of demand (e.g. shift to winter, overnight peaks) in ways that significantly affects the capacity value of resources (e.g. generally much lower for solar and better for wind, although it depends on the local wind patterns). (Note: France's greater seasonality discussed later in the same paragraph is due to prevalence of resistance electric heating btw. A presage of things to come with higher electrification of heating.)

p. 7, "Although dispatchable natural gas, biomass, or syngas generators equipped with 100%effective carbon capture storage (CCS) could provide system reliability without emissions2, underutilizedand capital-intensive back-up electricity would require higher investments and variable costs, and may beless efficient and effective when generators operate at low capacity factors." - Why would biomass or syngas plants (or hydrogen combustion turbines) have to run with CCS? These plants would be low capex and high fuel cost and thus well suited to this role of infrequent usage during 'dark doldrums' and other infrequent but prolonged periods of low wind/solar output.

Other resources like nuclear/advanced geothermal (which you dont mention) and gas or biomass w/CCS would have higher capex and lower fuel costs and suited to run at higher utilization rates. But this would just mean less solar and wind capacity needed (and storage). Cost optimized mixes would not try to max out wind and solar as an end in itself.

p. 7, "...second point has the effect of making our estimates for the efficacy of solar and wind resources to meet electricity demand conservative." - It makes the mean more conservative, but it probably reduces variance, as you have far more sites being averaged together. Please note both effects. The latter may actually be more important, since you are focused on reliability here (not economics), which concerns more the variability (rather than the mean capacity factor; a resource with low capacity factor but constant output would be more readily used to get to high levels of

reliability in your methods).

p. 7, discussion of dual axis trackers - What does dual axis tracking solar systems have to do with the use of area-weighted averages? Is this a distinct point? If so, start new paragraph or add clear transition paragraph. Not clear why this is here.

p. 7, "The feasibility of 12 or more hours of energy storage may depend on continued innovation and learning related to the associated materials and technologies." Here are a couple of more recent citations on longer duration storage:

<https://doi.org/10.1038/s41560-021-00796-8>

<https://doi.org/10.1016/j.joule.2019.06.012>

p. 7-8, discussion of sensitivity to load profile or lack thereof - I would be surprised if it were this insensitive to high levels of transport and heating electrification, which significantly shift the seasonal pattern and peakiness of demand. You could try using a demand pattern from the NREL Electrification Futures Study high electrification scenario to test this.

p. 8, "Despite these simplifying assumptions, our estimates of reliability compare favorably with the results of more detailed regional analyses that site generators and transmission more strategically and practically." - How do you support/justify this statement? What do you mean by "compare well" in this context? Seems like a vague assertion.

p. 8 "Our normalized analysis of the reliability for purely solar and wind supplied electricity system would apply as well to a system with other slowly time-varying generation (e.g. coal, hydro, geothermal, or nuclear) because the variability of solar and wind generation will have to be dealt with in either case to meet hourly demand. - Not clear what is meant by this. If there's sufficient less flexible firm generation capacity to meet the energy gaps shown in Fig 4, then reliability can be 100%, but at cost of additional wind/solar curtailment due to minimum output constraints for these generators. Again, this may not be cost-optimal, but neither are any of your scenarios here.

Reviewer #2 (Remarks to the Author):

First of all, thank you for the possibility to be one of the first readers of this undoubtedly interesting work. Here are some of my minor comments, which you might find relevant in improving the quality of your manuscript:

1) I am concerned with the process of obtaining/generating energy demand time series. It is highly unlikely that there is a single country for which hourly data is available for 39 years. But regardless of the above... the approach you have used... does it take into account the correlation between solar/wind availability and the load? It is clear (for example from ENTSO-E data) that there are some countries where we observe a stronger positive correlation between PV and demand - hence higher self-consumption. The same could be also observed potential for wind and heating systems. Can you comment on this?

2) Have you modelled some specific energy storage technology? 90% round-trip efficiency is most likely representative for batteries. What about the self-discharge?

3) I have to admit that your power supply model is very simple in its nature. Could you comment on how using a more sophisticated model which considers the cost of storage and electricity would your results change?

4) For wind power calculation you have used wind profile power law. Could you provide information about the alpha exponent that you have used?

5) Legend to Figure 1 could be larger in terms of font size.

6) The concept of complementarity is mentioned in your manuscript few times but its definition is

not provided. Simultaneously statements that something has a complementary nature or not is only supported by figures (a very good ones I have to admit). Solar Energy A review on the complementarity... is a paper giving a good overview on this concept and provided a bunch of useful metrics.

6) I think it would be beneficial to discuss for selected countries in more details the capacities (GW, GWh) that you are considering - just to give the reader an understanding what would be required for a reliable solar-wind-storage systems.

Summarizing. Very good and interesting work! Congratulations!

Reviewers' comments:

Reviewer #1 (Comments for the Author):

Reviewer #1 (Remarks to the Author):

Like the earlier work, Shaner et al., this is an interesting and potentially insightful contribution to the literature and I would welcome publication. However, I have three main/important methodological (or perhaps philosophical) questions about the work.

Response: We thank the Reviewer for his generally positive appraisal of our work and for the fair and constructive comments below. We've made a number of revisions in response and believe the manuscript has been substantially improved. Point-by-point responses are offered below.

1. Most importantly, are you really exploring/revealing any truly any fundamental geophysical constraints here? Isn't it a matter of economics in the end that will constraint (or land area/siting challenges that may bind even before economics)? So long as there are no hours with both zero solar and zero wind output, it should be mathematically possible to create a convex combination of wind and solar capacities that meet demand in all hours. If you add storage to the mix, then this gets easier. Even in countries with one or more hours of zero combined wind and solar output (e.g. a becalmed night), storage capacity can shift output from other periods to this time and achieve supply/demand balance. So what is the geophysical constraint or limit? There isn't really one, is there? You are revealing geophysical *considerations* or *dynamics* inherent in the temporal variability of wind and solar at different locations and spatial scales and across interannual variability. This is insightful in and of itself, but the real limits or constraints on renewables-dominated systems come from the economic and/or socio-political limitations that arise, in part, from these geophysical dynamics. (I had the same thoughts upon reading Shaner et al., and this new work raises them all again).

Response: We thank the Reviewer for the thoughtful comments. Reading the comment carefully, any disagreement we have relates to the strength of the word "constraint." By constraint, we do not mean to suggest an absolute limit that determines whether it is possible to meet electricity demand, but rather a critical factor that in turn may make reliable systems economically or socio-politically infeasible. The Reviewer acknowledges that relationship (e.g., "...economic and/or socio-political limitations that arise, in part, from these geophysical dynamics"), but interprets "constraint" as hard limit instead of an important restriction. In response to the comment, and recognizing that other readers might interpret "constraint" in this stronger sense but lacking a better word, we have added a sentence early in the text (Lines 43-45) that explains what we intend by "constraint" (Lines 43-45).

More particularly, we agree the Reviewer it's mathematically possible to create a convex combination of wind and solar capacities that meet demand in all hours, but, for example, our results show that the total capacity needed to meet a given country's demand in all hours increases exponentially after a point that depends on the renewable resources of that country, and it is this geophysically-dependent point that will largely determine the cost-effectiveness

of highly-reliable, renewables-based electricity systems. As noted above, we have revised the main text to reflect these points more clearly and hope that it allays the Reviewer's concerns (Lines 196-201).

“Note that we do not mean to suggest that the temporal variability of such resources would ever make it physically impossible to meet a given electricity demand, but rather the extent to which such variability may determine the economic or socio-political feasibility of reliable systems.”

“Although with vast enough wind and solar capacity it might still be possible to meet demand in all hours, the required capacity increases exponentially after a point that depends on the renewable resources of that country, and it is this geophysically-dependent point that will largely determine the cost-effectiveness of highly-reliable, renewables-based electricity systems.”

2. How do you validate your time series data (wind and solar simulations) and correct for biases in the hourly profiles of wind (and solar if relevant) in the reanalysis data? Reanalysis data series are known to have some challenges capturing in particular diurnal patterns of wind power output in some regions. The methods discussion mentions trying to correct annual capacity factors, which is a good step. But what about temporal patterns? As this is your focus here (hourly resolution variability in wind and solar series), this seems key, not just adjustments to correct bias in annual capacity factors.

Response: This is a good point, and we agree that hourly variations in the reanalysis data we used (MERRA-2) may be biased. In the revised manuscript, we have added new analyses based on a different and independent reanalysis product, ERA5, and compared the results. The somewhat different patterns of resource variability in the two datasets do not alter our main conclusions (see Supplementary Text 3 and Supplementary Figures S9-S10).

As shown in Supplementary Figure 9, our estimates of the system reliabilities by using ERA5 data in the 42 major countries are in good agreement with results of MERRA-2: under 1x generation and the most reliable mixes without storage, reliability under the different loads varies on average from -9.4% to 1.3% (see Fig. S9a). The differences are similar in systems with excess generation (Figs. S9).

We also compared the magnitude and duration of unmet demand in 16 major countries like Figure 4 (see Supplementary Figure 10). The data products of MERRA-2 and ERA5 both can essentially capture the number of hours each year that such a gap occurred. By contrast, the MERRA-2 data has a better performance of meeting hourly demand in larger countries (i.e. Russia and Canada) but a similar performance in small countries (i.e. United Kingdom).

Supplementary Figure 9 | The impacts of different reanalysis data products (MERRA-2 and ERA5) on the system reliability in the 42 major countries. generation quantities are varied in each panel: (a) 1 x generation without storage; (b) 1.5 x generation without storage; (c) 3x generation without storage. The values mean the reliability change ratio (%) of 2-axis solar tracking system comparing to single axis solar tracking system.

Supplementary Figure 10 | Average power supply gap comparison. The power supply gap comparison of the most reliable electricity system are shown by generation quantity by using reanalysis data MERRA-2 and ERA5: (a) Russia; (b) Canada; (c) contiguous U.S.; (d) China; (e) Brazil; (f) Australia; (g) India; (h) Algeria; (i) Mexico; (j) South Africa; (k) France; (l) Japan; (m) Germany; (n) New Zealand; (o) United Kingdom; (p) South Korea.

3. I would expect that the marginal reliability benefit of increased capacity of storage or increased overbuild of wind/solar declines steadily (and perhaps rapidly, based on earlier works on these topics). You do not explore these marginal dynamics at all in your work, which I would think would be one of the main benefits of taking a simplified approach as in this study (without cost optimization): you can trace out marginal changes across varying levels of overbuild and/or storage capacity (perhaps two-d or isoquant plots showing both dimensions). You present some of these factors as if they are fixed/constant, but that is unlikely. For example, p. 5 states "Our results indicate that a 10% increase in excess annual generation is equivalent to 3.9 hours of storage" but is this substitution ratio consistent? Does it vary across nations and at different overbuild levels? I would expect this to change on the margin, so first 10% excess

generation might be worth 3.9 hours, but next 10% would be worth less, etc. There should be diminishing marginal returns here.

Response: We thank the Reviewer for the constructive comments. Our estimates indeed show that the marginal reliability benefit of increased capacity of storage or increased overbuild of wind/solar declines steadily. Under a given capacity of energy storage (e.g., 3 hours), our results of 1x, 1.5x, and 3x generation shows that the first 10% excess generation increase is larger than the next 10% excess generation increase (i.e. the marginal benefit for system reliability decreases as excess generation increases).

As might be expected, the diminishing marginal benefits between excess generation and increased storage apply in both directions. Our fitting model performs well across different nations, overbuild levels, and storage levels. The differences in reliability between the estimates and the model predicted values are between -5.5% to 5.8% and ~ 80% of the differences are within $\pm 2\%$, with no systematic bias related to region or the magnitude of overbuild or storage. Nonetheless, our model and conclusions are limited by our experimental design and the discrete levels of excess generation (1x, 1.5x, and 3x) and storage (0h, 3h, and 12h) we evaluated.

We have added the discussion of these diminishing returns/marginal benefits as suggested (Lines 245-255).

“Our estimates show that the marginal reliability benefit of increased capacity of storage or increased overbuild of wind/solar declines steadily. Under a given capacity of energy storage (e.g., 3 hours), our results of 1x, 1.5x, and 3x generation shows that the first 10% excess generation increase is larger than the next 10% excess generation increase (i.e. the marginal benefit for system reliability decreases as excess generation increases). As might be expected, the diminishing marginal benefits between excess generation and increased storage apply in both directions. Our fitting model performs well across different nations, overbuild levels, and storage levels. The differences in reliability between the estimates and the model predicted values are between -5.5% to 5.8% and ~80% of the differences are within $\pm 2\%$, with no systematic bias related to region or the magnitude of overbuild or storage. Nonetheless, our model and conclusions are limited by our experimental design and the discrete levels of excess generation (1x, 1.5x, and 3x) and storage (0h, 3h, and 12h) we evaluated.”

-Signed,
Jesse D. Jenkins

These are my major/substantive concerns with the work.

Specific comments follow:

p. 1, abstract: "If future net-zero emissions energy systems rely heavily on solar and wind resources, spatial and temporal mismatches between resource availability and electricity demand may degrade system reliability." Add "or require installation of sufficient firm generation capacity and/or long-duration energy storage."

Response: Thanks for the constructive suggestion, and we have revised as suggested.

p. 1, "in most scenarios" - revise to "most cost-optimized scenarios"?

Response: Revised as suggested.

p. 1 "The efficacy of meeting electricity demands with generation from solar and wind resources depends on factors such as location and weather; the area over which generating assets are distributed; the mix and magnitude of solar and wind generation capacities; and the availability of energy storage" - insert "and firm generation capacity." Consider adding cite to <https://doi.org/10.1016/j.joule.2018.08.006> or similar work on firm generation.

Response: We have revised and added the reference (Sepulveda et al., 2018) as suggested.

Reference:

Sepulveda, N. A., Jenkins, J. D., de Sisternes, F. J. & Lester, R. K. The Role of Firm Low-Carbon Electricity Resources in Deep Decarbonization of Power Generation. *Joule* 2, 2403–2420 (2018).

p. 1, "Resource adequacy planning standards are also high: in North America (BAL- 502-RF-03)17, generating resources must be adequate to provide no more than 1 day of unmet electricity demand in 10 years (i.e. 99.97%)." Note that different jurisdictions define 1 in 10 standard differently. Some define it as one day in 10 years (so 99.97%) but some define it as one loss of load event (of unspecified duration, as short as one hour) per 10 years, which implies generally higher reliability (closer to '4 nines' or 99.99%). See <https://pubs.naruc.org/pub/FA865D94-FA0B-F4BA-67B3-436C4216F135#:~:text=As%20recognized%20in%20the%20recent,relability%20standard%20has%20different%20interpretations.&text=one%20event%20in%2010%20years,LOLE%20in%20events%20per%20year>.

Response: We thank the Reviewer for the constructive comment, and we have added the description of different “1-in-10 standard” in the revised main text and corresponding reference as suggested.

“Resource adequacy planning standards for “1-in-10” are also high: in North America (BAL-502-RF-03), generating resources must be adequate to provide no more than 1 day of unmet electricity demand—or in some cases 1 loss of load event—in 10 years (i.e. 99.97% or 99.99%, respectively)”

Reference:

Carden. K., Wintermantel, N. & Pfeifenberger J. The Economics of Resource Adequacy Planning: Why Reserve Margins Are Not Just About Keeping the Lights On, 2011 National Regulatory Research Institute.

p. 1, "Our goal is to identify the opportunities and challenges of variable renewable resources

in greater detail than can be done by integrated assessment models that have multi- year time steps." How is this work situated relative to cost-optimized capacity expansion planning models with hourly chronological operating detail (e.g. Switch, Pypsa, GenX, urbs, OpenCEM, etc)? And why is your approach the appropriate one for this purpose? I'd also suggest expanding this first comparison point to both IAMs and energy-economic models like NEMS or TIMES/MARKAL, as your are equally valid in your comparison to both here.

Response: We thank the Reviewer for the constructive comments. Indeed, our results offer insights over both multi-year time step IAMs and other hourly energy system models. Specifically, its hourly resolution and several decades long timespan allow us to evaluate the reliability implications of regional differences in solar and wind resources independent of costs, which marks a clear advance over IAMs but also helps to explain the results of cost-optimized models like those pointed out by the Reviewer. In particular, many of the cost-optimized models are used to generate scenarios in which either renewables sources are required to meet a very high share of demand or else renewables are assumed to be extremely cheap but nonetheless indicate substantial increases in system costs related to, e.g., energy storage. Our geophysically-focused results help to explain such results irrespective of cost assumptions. We have added the clarification in the revised text (Lines 288-301).

In response to the Reviewer's comment, we have expanded the text comparing to other models and explaining the relative value of our approach, and have added new references for several techno-economic studies that have used independent approaches to model U.S. solar- and wind-dominated electricity systems in detail. In each case, these studies find that the share of non-emitting (or carbon neutral) electricity contributed by solar and wind in cost-optimized systems is rarely in excess of 80%, with the residual demand for non-emitting generation met by firmer renewables such as biomass, hydroelectricity, and geothermal (Larson et al., 2020; Williams et al., 2021; MacDonald et al., 2016).

"Despite these simplifying assumptions, our results offer insights over both multi-year time step integrated assessment modes (IAMs) and other hourly energy system models. Specifically, its hourly resolution and several decades long timespan allow us to evaluate the reliability implications of regional differences in solar and wind resources independent of costs, which marks a clear advance over IAMs but also helps to explain the results of cost-optimized models. In particular, many of the cost-optimized models are used to generate scenarios in which either renewables sources are required to meet a very high share of demand or else renewables are assumed to be extremely cheap but nonetheless indicate substantial increases in system costs related to, e.g., energy storage. Our geophysically-focused results help to explain such results irrespective of cost assumptions. We compare the estimates of reliability and capacities in this study with several techno-economic studies that have used independent approaches to model regional solar- and wind-dominated electricity systems in detail. In each case focusing U.S., these studies find that the share of non-emitting (or carbon neutral) electricity contributed by solar and wind in cost-optimized systems is typically ~80%, with the residual demand for non-emitting generation met by firmer renewables such as biomass, hydroelectricity, and geothermal."

References:

E. Larson, C. Greig, J. Jenkins, E. Mayfield, A. Pascale, C. Zhang, J. Drossman, R. Williams, S. Pacala, R. Socolow, EJ Baik, R. Birdsey, R. Duke, R. Jones, B. Haley, E. Leslie, K. Paustian, and A. Swan, Net-Zero America: Potential Pathways, Infrastructure, and Impacts, interim report, Princeton University, Princeton, NJ, December 15, 2020.

Williams, J. H. *et al.* Carbon-Neutral Pathways for the United States. *AGU Advances* **2**, e2020AV000284 (2021).

MacDonald, A. E. *et al.* Future cost-competitive electricity systems and their impact on US CO₂ emissions. *Nature Clim Change* **6**, 526–531 (2016).

p. 2, "the great electricity demand" - should say "greatest" not "great"

Response: Revised as suggested.

p. 2, "Electricity demand profiles for each country are determined by factors such as economic conditions, prevailing weather conditions and consumer usage patterns" - Note for the reader that you are using historic demand patterns and are not capturing plausible changes in demand due to electrification, managed charging/flexible loads, or introduction of electrolysis or other large flexible demands in future years concurrent with high penetrations of renewables. Another important limitation to note. Electrification can dramatically change demand profiles, particularly in northern latitude countries not currently dependent on electric heat. This changes both the diurnal and seasonal patterns of demand (e.g. shift to winter, overnight peaks) in ways that significantly affects the capacity value of resources (e.g. generally much lower for solar and better for wind, although it depends on the local wind patterns). (Note: France's greater seasonality discussed later in the same paragraph is due to prevalence of resistance electric heating btw. A presage of things to come with higher electrification of heating.)

Response: We agree that changes in the temporal patterns of demand could substantially affect our results.

As suggested in a subsequent comment by the Reviewer, we have performed new analyses using the demand pattern from the high electrification scenario in the NREL Electrification Futures Study and use the results to discuss the sensitivity of our results to such different load profiles.

For example, the results of this test for the U.S. show that reliability is not especially sensitive to the high electrification demand profile: under 1x generation without storage, reliability under the different renewable mixes varies on average from -1% to 2.5% (see Supplementary Fig. S11). The differences are even smaller in solar-heavy systems with excess generation.

Supplementary Figure 11 | The impacts of future high electrification demand profile on the system reliability in the U.S..

Reference:

Mai, Trieu, Paige Jadun, Jeffrey Logan, Colin McMillan, Matteo Muratori, Daniel Steinberg, Laura Vimmerstedt, Ryan Jones, Benjamin Haley, and Brent Nelson. 2018. Electrification Futures Study: Scenarios of Electric Technology Adoption and Power Consumption for the United States. National Renewable Energy Laboratory. NREL/TP-6A20-71500. <https://doi.org/10.2172/1459351>.

p. 7, "Although dispatchable natural gas, biomass, or syngas generators equipped with 100% effective carbon capture storage (CCS) could provide system reliability without emissions², underutilized and capital-intensive back-up electricity would require higher investments and variable costs, and may be less efficient and effective when generators operate at low capacity factors." - Why would biomass or syngas plants (or hydrogen combustion turbines) have to run with CCS? These plants would be low capex and high fuel cost and thus well suited to this role of infrequent usage during 'dark doldrums' and other infrequent but prolonged periods of low wind/solar output.

Response: We thank the Reviewer for the comment. The sentence should not have combined natural gas (which would need to be equipped with CCS) and the other fuels (assuming they are carbon neutral). However, although the Reviewer is quite right that combustion turbines or combined cycle plants burning biogas, syngas, or hydrogen might have low capex, requisite industrial-scale facilities for biodigestion, direct air capture, Fischer-Tropsch, and/or electrolysis will not be. Sector-coupling or right-sizing of these fuel-production facilities could

nonetheless make infrequent operation of generators feasible (Dowling et al. 2020). In response to the Reviewer’s comment, we have clarified these points in the revised text (Lines 201-206).

“In contrast, combustion turbines or combined cycle plants burning carbon-neutral biogas, syngas, or hydrogen might have comparatively low capital costs, but would require additional and large capital investments to produce such fuels (e.g., biodigestion, direct air capture, Fischer-Tropsch, and/or electrolysis). Sector-coupling or right-sizing of these net-zero emissions fuel-production facilities could nonetheless make infrequent operation of generators feasible.”

Reference:

Dowling, J. A. et al. Role of Long-Duration Energy Storage in Variable Renewable Electricity Systems. *Joule* 4, 1907–1928 (2020).

Other resources like nuclear/advanced geothermal (which you dont mention) and gas or biomass w/CCS would have higher capex and lower fuel costs and suited to run at higher utilization rates. But this would just mean less solar and wind capacity needed (and storage). Cost optimized mixes would not try to max out wind and solar as an end in itself.

Response: We agree with the Reviewer that more firm generation would mean less solar and wind capacity in a given system, and of course cost-optimization will determine the technological composition of a system based on assumed costs. But as the Reviewer knows well, many jurisdictions and advocates are indeed interested in “maxing out” solar and wind. The purpose of our study is thus to explore the implications of country-level differences in the variability of solar and wind resources, including how much storage and firm generation might be required to ensure resource adequacy. We do not analyze which specific energy techs would actually provide such storage or firm generation.

Although methods are simple and transparent, our goals and findings are remarkably consistent with much more complex approaches. For example, the recently published Net-Zero America report includes a cost-optimized “all-renewables” scenario which decarbonizes U.S. electricity without nuclear or CCS: by 2050, ~81.6% of primary energy in the E+RE+ scenario is from solar and wind (Larson et al. 2020).

The revised text discusses these points (Lines 206-214).

“More firm generation would mean less solar and wind capacity in a given system, which might or might not be cost-effective depending on technology costs. But many jurisdictions and advocates are interested in “maxing out” solar and wind. Our results are especially relevant in that context, highlighting the implications of country-level differences in the variability of solar and wind resources, including how much storage and firm generation might be required to ensure resource adequacy. Although methods are simple and transparent, our goals and findings are remarkably consistent with much more complex approaches. For example, the recently published Net-Zero America report includes a cost-optimized “all-renewables” scenario which decarbonizes U.S. electricity without nuclear or CCS: by 2050, ~81.6% of

primary energy in the E+RE+ scenario is from solar and wind.”

Reference:

E. Larson, C. Greig, J. Jenkins, E. Mayfield, A. Pascale, C. Zhang, J. Drossman, R. Williams, S. Pacala, R. Socolow, EJ Baik, R. Birdsey, R. Duke, R. Jones, B. Haley, E. Leslie, K. Paustian, and A. Swan, Net-Zero America: Potential Pathways, Infrastructure, and Impacts, interim report, Princeton University, Princeton, NJ, December 15, 2020.

p. 7, "...second point has the effect of making our estimates for the efficacy of solar and wind resources to meet electricity demand conservative." - It makes the mean more conservative, but it probably reduces variance, as you have far more sites being averaged together. Please note both effects. The latter may actually be more important, since you are focused on reliability here (not economics), which concerns more the variability (rather than the mean capacity factor; a resource with low capacity factor but constant output would be more readily used to get to high levels of reliability in your methods).

Response: We thank the Reviewer for the constructive comments. This could be limitation and we have added possible concerns as suggested (Lines 230-232).

“However, either method to calculate capacity factors of national and regional area-weighted averages may also reduce the resource variability and thereby increase estimates of reliability.”

p. 7, discussion of dual axis trackers - What does dual axis tracking solar systems have to do with the use of area-weighted averages? Is this a distinct point? If so, start new paragraph or add clear transition paragraph. Not clear why this is here.

Response: We apologize for the confusion. The calculation of solar capacity factor for dual-axis tracking solar system does not involve the use of area-weighted averages. However, since different grid cells in the reanalysis product have different grid areas, the area-weighted averaging process is used when calculating the continental-scale hourly solar capacity factor. And the mentioned dual axis tracking solar systems here is to investigate the impacts of estimated solar capacity factors from different methods (i.e. horizontal single-axis and dual-axis solar tracking systems) on the system reliability (see Figure S8). We have clarified this in the revised text (Lines 224-227).

“This second point has the effect of making our estimates for the efficacy of solar and wind resources to meet electricity demand more conservative by using the horizontal single-axis tracking system compared to the dual-axis solar tracking systems.”

p. 7, "The feasibility of 12 or more hours of energy storage may depend on continued innovation and learning related to the associated materials and technologies." Here are a couple of more recent citations on longer duration storage:

<https://doi.org/10.1038/s41560-021-00796-8>

<https://doi.org/10.1016/j.joule.2019.06.012>

Response: Thank you for pointing these out. We agree they're very relevant and have added them as suggested.

References:

Sepulveda, N. A., Jenkins, J. D., Edington, A., Mallapragada, D. S. & Lester, R. K. The design space for long-duration energy storage in decarbonized power systems. *Nat. Energy* (2021) doi:10.1038/s41560-021-00796-8.

Ziegler, M. S. et al. Storage Requirements and Costs of Shaping Renewable Energy Toward Grid Decarbonization. *Joule* **3**, 2134–2153 (2019).

p. 7-8, discussion of sensitivity to load profile or lack thereof - I would be surprised if it were this insensitive to high levels of transport and heating electrification, which significantly shift the seasonal pattern and peakiness of demand. You could try using a demand pattern from the NREL Electrification Futures Study high electrification scenario to test this.

Response: We really appreciate this constructive suggestion. As mentioned above, we have performed additional analyses using the demand pattern in the NREL study as suggested. When combining the high electrification scenario and rapid technology advancement for the U.S., our results show that reliability is not especially sensitive to demand profile: under 1x generation without storage, reliability under the different mixes varies on average from -1% to 2.5% (see Fig. S11 above).

Reference:

Mai, Trieu, Paige Jadun, Jeffrey Logan, Colin McMillan, Matteo Muratori, Daniel Steinberg, Laura Vimmerstedt, Ryan Jones, Benjamin Haley, and Brent Nelson. 2018. Electrification Futures Study: Scenarios of Electric Technology Adoption and Power Consumption for the United States. National Renewable Energy Laboratory. NREL/TP-6A20-71500. <https://doi.org/10.2172/1459351>.

p. 8, "Despite these simplifying assumptions, our estimates of reliability compare favorably with the results of more detailed regional analyses that site generators and transmission more strategically and practically." - How do you support/justify this statement? What do you mean by "compare well" in this context? Seems like a vague assertion.

Response: We agree the original assertion was overly vague. We have added references in the revised text to several techno-economic studies that have used independent approaches to model U.S. solar- and wind-dominated electricity systems in detail. In each case, these studies find that the share of non-emitting (or carbon neutral) electricity contributed by solar and wind in cost-optimized systems is rarely more than ~80%, with the residual demand for non-emitting generation met by firmer renewables such as biomass, hydroelectricity, and geothermal (Larson et al., 2020; Williams et al., 2021; MacDonald et al., 2016).

“We compare the estimates of reliability and capacities in this study with several techno-economic studies that have used independent approaches to model regional solar- and wind-dominated electricity systems in detail. In each case focusing U.S., these studies find that the share of non-emitting (or carbon neutral) electricity contributed by solar and wind in cost-optimized systems is typically ~80%, with the residual demand for non-emitting generation met by firmer renewables such as biomass, hydroelectricity, and geothermal”

References:

E. Larson, C. Greig, J. Jenkins, E. Mayfield, A. Pascale, C. Zhang, J. Drossman, R. Williams, S. Pacala, R. Socolow, EJ Baik, R. Birdsey, R. Duke, R. Jones, B. Haley, E. Leslie, K. Paustian, and A. Swan, Net-Zero America: Potential Pathways, Infrastructure, and Impacts, interim report, Princeton University, Princeton, NJ, December 15, 2020.

Williams, J. H. *et al.* Carbon-Neutral Pathways for the United States. *AGU Advances* **2**, e2020AV000284 (2021).

MacDonald, A. E. *et al.* Future cost-competitive electricity systems and their impact on US CO₂ emissions. *Nature Clim Change* **6**, 526–531 (2016).

p. 8 "Our normalized analysis of the reliability for purely solar and wind supplied electricity system would apply as well to a system with other slowly time-varying generation (e.g. coal, hydro, geothermal, or nuclear) because the variability of solar and wind generation will have to be dealt with in either case to meet hourly demand. - Not clear what is meant by this. If there's sufficient less flexible firm generation capacity to meet the energy gaps shown in Fig 4, then reliability can be 100%, but at cost of additional wind/solar curtailment due to minimum output constraints for these generators. Again, this may not be cost-optimal, but neither are any of your scenarios here.

Response: We are again sorry for the confusion and have revised the related text as suggested. What we mean is that renewable variability must be managed regardless of back-up energy sources, so that either renewable generation is curtailed or firm capacity idled (except where coupled to other sectors). In the revised text we clarify that our discussion is not about cost-optimal systems (Lines 315-317).

“Our normalized analysis of the reliability for purely solar and wind supplied electricity system would apply as well to a system with other slowly time-varying generation (e.g. coal, hydro, geothermal, or nuclear) because the variability of solar and wind generation is always challenge in either back-up-sources-mixed case to meet hourly demand within the 100% reliability power system, which will have to be managed and dealt with when turning back-up sources up and down or curtailing excess solar and wind.”

Reviewer #2 (Comments for the Author):

Reviewer #2 (Remarks to the Author):

First of all, thank you for the possibility to be one of the first readers of this undoubtedly interesting work. Here are some of my minor comments, which you might find relevant in improving the quality of your manuscript:

Response: We thank the Reviewer for the positive tone of our work and for the fair and constructive comments below. We've made a number of revisions in response and believe the manuscript has been substantially improved. Point-by-point responses are offered below.

1) I am concerned with the process of obtaining/generating energy demand time series. It is highly unlikely that there is a single country for which hourly data is available for 39 years. But regardless of the above... the approach you have used... does it take into account the correlation between solar/wind availability and the load? It is clear (for example from ENTSO-E data) that there are some countries where we observe a stronger positive correlation between PV and demand - hence higher self-consumption. The same could be also observed potential for wind and heating systems. Can you comment on this?

Response: This is an excellent comment. Electricity demand may be correlated with the availability of resources, for example peak cooling loads could correspond to sunny but stagnant (low wind) days. However, our analytical approach will not be affected by such covariance because the supply and demand are not from the same years: we analyze each of the 39 years of resource data against a single recent year of demand. Moreover, our analysis aggregates both supply and demand over entire countries which would attenuate any such correlations occurring at smaller geographical scales. Challenges posed by normal seasonal and diurnal patterns are nonetheless captured.

We analyzed our dataset but didn't find stronger positive correlation between solar availability and demand at national scale. We have added some discussions of this comment in the revised text (Lines 271-275).

“For example, stronger positive correlation between solar/wind availability and demand may be observed as renewable energy gradually dominates the power system. However, our analysis compares resources and demand in different years and at the country-level, which should preclude any bias related to specific subnational weather events.”

2) Have you modelled some specific energy storage technology? 90% round-trip efficiency is most likely representative for batteries. What about the self-discharge?

Response: Storage in our model operates with 90% round-trip efficiency with self-discharge of 1% per month, reflecting the high-end performance of current batteries (see, e.g., Pellow et al., 2015 and Wang et al., 2018). In response to the Reviewer's comment, we have clarified these details in the revised text (Lines 414-415).

References:

Pellow, M.A., Emmott, C.J., Barnhart, C.J. & Benson, S.M. Hydrogen or batteries for grid storage? A net energy analysis. *Energy Environ. Sci.* **8**, 1938–1952 (2015).

Wang, H., Jiang, Y. & Manthiram, A. Long Cycle Life, Low Self-Discharge Sodium–Selenium Batteries with High Selenium Loading and Suppressed Polyselenide Shuttling. *Adv. Energy Mater.* **8**, 1701953 (2018).

“We also assumed a storage charging round-trip efficiency and storage decay rate of respectively 90% and 1.14×10^{-6} per hour (i.e. 1% of stored electricity lost per month), reflecting the high-end performance of current batteries.”

3) I have to admit that your power supply model is very simple in its nature. Could you comment on how using a more sophisticated model which considers the cost of storage and electricity would your results change?

Response: The Reviewer is of course correct that our model is quite simple; we think its simplicity makes it easier to understand and interpret our results. We think they are quite complementary to cost-optimized model results where it is difficult to understand whether the resources or underlying cost assumptions are driving their results. In response to the Reviewer’s comment and those of Reviewer #1, we have added discussion and references to several independent techno-economic studies that have modeled solar- and wind-dominated electricity systems for the U.S. in detail. In each case, these studies find that the share of non-emitting (or carbon neutral) electricity contributed by solar and wind in cost-optimized systems is rarely more than ~80%, with the residual demand for non-emitting generation met by firmer renewables such as biomass, hydroelectricity, and geothermal (Larson et al., 2020; Williams et al., 2021; MacDonald et al., 2016).

“Our geophysically-focused results help to explain such results irrespective of cost assumptions. We compare the estimates of reliability and capacities in this study with several techno-economic studies that have used independent approaches to model regional solar- and wind-dominated electricity systems in detail. In each case focusing U.S., these studies find that the share of non-emitting (or carbon neutral) electricity contributed by solar and wind in cost-optimized systems is typically ~80%, with the residual demand for non-emitting generation met by firmer renewables such as biomass, hydroelectricity, and geothermal.”

References:

E. Larson, C. Greig, J. Jenkins, E. Mayfield, A. Pascale, C. Zhang, J. Drossman, R. Williams, S. Pacala, R. Socolow, EJ Baik, R. Birdsey, R. Duke, R. Jones, B. Haley, E. Leslie, K. Paustian, and A. Swan, Net-Zero America: Potential Pathways, Infrastructure, and Impacts, interim report, Princeton University, Princeton, NJ, December 15, 2020.

Williams, J. H. *et al.* Carbon-Neutral Pathways for the United States. *AGU Advances* **2**, e2020AV000284 (2021).

MacDonald, A. E. *et al.* Future cost-competitive electricity systems and their impact on US CO₂ emissions. *Nature Clim Change* **6**, 526–531 (2016).

4) For wind power calculation you have used wind profile power law. Could you provide information about the alpha exponent that you have used?

Response: The alpha exponent for wind profile calculation is estimated based on the 10m and 50m wind speed for each grid cell using the following equation:

$$\alpha = \frac{\log(50m \text{ wind speed}) - \log(10m \text{ wind speed})}{\log(50) - \log(10)}$$

Then the 100m height wind speed is calculated as:

$$100m \text{ wind speed} = 10m \text{ wind speed} * \left(\frac{100}{10}\right)^\alpha$$

We have added above information in the revised main text (Lines 352-357).

5) Legend to Figure 1 could be larger in terms of font size.

Response: Revised as suggested (also revised Supplementary Figure 1).

6) The concept of complementarity is mentioned in your manuscript few times but its definition is not provided. Simultaneously statements that something has a complementary nature or not is only supported by figures (a very good ones I have to admit). Solar Energy A review on the complementarity... is a paper giving a good overview on this concept and provided a bunch of useful metrics.

Response: We thank the Reviewer for the constructive comments. We have added the definition of complementarity in the revised main text and also added reference as suggested.

In addition, Kendall correlation coefficient for energetic complementarity assessment is chosen and supplemented in this study to assess the 39-years temporal characteristics of wind and solar complementarity in 42 main counties (see Supplementary Table 2). And the Kendall's correlation coefficients of 42 main countries are -0.91~-0.83, which means the good complementarity between solar and wind resources because the value of -1 indicates the best possible complementarity.

“The complementarity of renewable energy sources for this study is defined as a hybridization of solar-wind resources over over a given area (here, countries), which we estimate by the Kendall correlation coefficient of these resources across 39-years of resource data.”

“Kendall's correlation coefficients of solar and wind resources in the 42 main countries range from -0.91 to -0.83 (see Supplementary Table 2), another indication of good complementarity (where -1 is the best possible complementarity).”

Reference:

Jurasz, J., Canales, F. A., Kies, A., Guezgouz, M. & Beluco, A. A review on the complementarity of renewable energy sources: Concept, metrics, application and future research directions. *Solar Energy* 195, 703–724 (2020).

6) I think it would be beneficial to discuss for selected countries in more details the capacities (GW, GWh) that you are considering - just to give the reader an understanding what would be required for a reliable solar-wind-storage systems.

Response: We thank the Reviewer for the constructive comments. We have created a new summary table with this information and have added it to the revised Supporting Information (see Supplementary Table 5).

Summarizing. Very good and interesting work! Congratulations!

Response: We thank the Reviewer for the positive tone of our work again.

REVIEWERS' COMMENTS

Reviewer #1 (Remarks to the Author):

The authors have done a commendable job responding thoroughly and substantively to all points raised in my review. Additional analysis and revisions throughout address all concerns, and the paper is stronger for it. I look forward to seeing and sharing it in publication.

-Jesse Jenkins

Reviewer #2 (Remarks to the Author):

Thank you for the revisions and provided responses. I find them satisfactory. The article is an important contribution in the field of energy research.

Reviewers' comments:

Reviewer #1 (Remarks to the Author):

The authors have done a commendable job responding thoroughly and substantively to all points raised in my review. Additional analysis and revisions throughout address all concerns, and the paper is stronger for it. I look forward to seeing and sharing it in publication.

-Jesse Jenkins

Response: We thank the careful review from the Referee and we appreciate that the Referee is satisfied with our revisions.

Reviewer #2 (Remarks to the Author):

Thank you for the revisions and provided responses. I find them satisfactory. The article is an important contribution in the field of energy research.

Response: We thank the careful review from the Referee and we appreciate that the Referee is satisfied with our revisions.